# Observer-dependent Collective Behavior For Biologically-inspired Processing Models

**Anonymous Author(s)**\*

## Abstract

The role of observers in computational learning models is rarely discussed, but offers a unique perspective on the practice of analysis and holistic interpretation of data. In this paper, we present an observer-dependent perspective of data acquisition and preprocessing. Observers consist of observers which exhibit environmental relativism, biorealism, and collective behavior. Environmental realism is achieved by introducing a variety of viewpoints to a population of observers, while biorealism provides observers that have an innate component that resembles a genotype. Collectively, observers can enable emergent features in a given dataset, as well as contextual understanding. The individual and collective behavior of observers can serve as a preprocessing layer or pretrained model for a variety of machine and deep learning models. In considering the collective behavior of our computational observers, we make a number of predictions about their behavior that may facilitate specific applications.

## 1 Introduction

When data is sampled for analysis by computational and statistical models, the sampling strategy is usually passive. This hypothetical approach considers data points or environmental features in a serial and homogeneous manner. A serial sampling strategy refers to acquiring information from single points in time, whereas the homogeneous sampling strategy acquires information from an independent physical point-of-view. What is needed is an observer or an agent that exhibits self-awareness of its role in the actions in the system. Multi-agent algorithms are notable for what we might call an Active Sampling strategy: they sample the environment from many different perspectives and can produce a number of collective behaviors [1]. This example shows that Active Sampling combined with an explicit role for agents can improve situational context. While some agent-based models [2] explicitly define a role for the observer in computational models, consideration of how these agents might behave and what roles they might perform is needed. Perhaps more importantly, many models of machine learning treat the observer as an implicit component of the model. The fields of Explainable AI and Algorithmic Interpretability are an exception to this, and consider the user as a key aspect of algorithm design [3]. In some cases, models such as the explainable observer-classifier [4] are used, which formalizes a classification problem as the domain of two agents that communicate with each other.

These sampled perspectives are analyzed and synthesized by the observer's internal model, which consists of a network that contains a representation of the observer's genotype-to-phenotype (G-to-P) mapping. This representation is similar to the Part-of-Speech (PoS) tagger enabled by Particle Swarm Optimization as described in [5]. This mapping defines the observer's biological substrate or

---

\*

Preprint. Under review at the 2nd Shared Visual Representations in Human and Machine Intelligence (SVRHM) Workshop at NeurIPS 2020.

relationship between innate components (genotype) and a conceptual map (phenotype). The innate (or genotypic) representation consists of a genotype that exhibits genetic diversity. This diversity allows for a stochastic sampling heterogeneity that enables bottom-up phenotypic diversity. The phenotype also relies on a top-down mechanism that helps to select upon the G-to-P mapping. In general, top-down phenotypic diversity is introduced through stimulus diversity. Stimulus diversity is achieved by introducing a stimulus (in this case, geometric objects under transformation) from various perspectives.

Our effort extends this line of work in a number of innovative ways. We propose an architecture for computational agents (observers) situated in an environmental context consisting of various geometric perspectives. This allows for populations of observers to take multiple perspectives on a given dataset or scene. The proposed representation allows us to discuss two concepts related to data, observers, and their context: environmental relativism and biorealism. Using a G-to-P mapping provides a balance between individualistic behavior while also being able to specify a hard-coded set of behavioral traits.

## 1.1 Environmental Relativism, Biorealism, and the Observer

In this paper, we assume a set of heterogeneous observers will take different but ultimately complementary perspectives on common stimuli that are transformed in various ways. Our observers utilize a subset of these perspectives which yields a network model of shared context. We can use various transformations of images or geometric shapes as perspective data in a manner similar to modern data augmentation schemes [6]. These stimuli can exist as affine transformational perspectives in a real-world context, or deformed transformational perspectives in a virtual world context. We describe one possible implementation of differential stimulus perspectives in the Appendix.

In a more general setting, training observers on multiple perspectives has numerous advantages when understood as viewpoints. In a classic paper, Rouse [7] discusses the role of first- and third-person views in video gaming, particularly the importance of seeing objects and worlds from various perspectives. Incorporating different views of the same set of objects contributes to improved performance and social presence [8, 9]. Such perspectives require the rotation of angles, alignments, and paths of motion, and a diversity of these viewpoints have a significant impact on the cognitive states of the viewers [10]. This can also simulate cognitive processes such as motor learning [11].

We predict that these findings on viewpoints will also hold true for computational observers. For example, Ye et.al [12] used relativistic perspectives to introduce agents to a diverse set of inputs in an open-ended manner, with which they were able to generalize a deep learning model for one-shot learning. Additionally, the degrees of freedom introduced by transformations of image perspective contributes to the need for greater social presence. This can be understood in terms of computational complexity [13] and has been further demonstrated through the world design of video games such as Pac-Man [14], Minesweeper [15], and classic Nintendo games [16].

## 1.2 Predictions for a Observer Representation

We predict that the perspective diversity inherent in a heterogeneous set of observers will result in a preference for some perspectives over others, which can evolve on the basis of a generative internal model. This is based on a population genetics-inspired hybrid connectionist//GA model first described in Alicea [17]. While conventional connectionist models focus on pattern recognition and associative learning, our hybrid connectionist model takes a different approach. Our approach is to incorporate a genetic representation to produce an internal model that represents a simple behavioral phenotype. In this way, we can produce stereotypical behaviors that incorporate environmental feedback. Yet, more importantly, we can also produce phenotypic diversity across observers, which potentially provides a nuanced perspective on the study of behavior.

# 2 Methodological Approaches

## 2.1 Observers in (and of) Complex Systems

Understanding how the observer has been conceptualized in scientific inquiry is an important first step for appreciating observer-dependence. One set of examples of the observer comes from quantum

physics [18] and complex systems [19]. Another example comes from embodiment and enactivism [20]. Observers are also crucial in agent-based modeling [21], which is perhaps closest to the current work. In the realm of quantum physics [22], multiple observers of the same phenomenon are crucial in acquiring an objective measurement of decoherence (quantum state). The cybernetics perspective first advanced by Ashby [23] postulates that the observer interacts with a black box so that the observer can only access externally states emitted by a hidden internal process.

In many cases, however, the autonomous behavior of the observer is indistinguishable from interactions between the observer and the observed object [19]. One way around this is through a strategy called the redundant spreading of information [24]. Suppose a population of observers observe independent features collected in distinct parts of the environment being surveyed. This allows us to obtain an objective estimate of the true state of this environment in a manner similar to the Quantum Darwinism approach [25], where the most commonly observed features across all observers are considered to have the highest fitness.

## 2.2   G-to-P Mapping as a Route to Objectivity

In this paper, we provide a rough sketch of the G-to-P approach. The input layer of the G-to-P representation is based on a set of functionally-explicit loci (genomic sites) modeled using boolean strings. This is analogous to a series of genes composed of DNA bases. The internal nodes are boolean switches that model the graded regulation of each loci. The output layer is based on a phenotype that encounters sensory information including visual, haptic, and thermal features. Environmental relativism allows us to think about data from intentionally different perspectives (circumstances afford differing viewpoints), while biorealism allows us to conceptualize the observer as an organism that has an innate component [26]. As our examples from quantum physics suggest, our aim is to bring us closer to objectivity through subjective evaluation. An observer can also compare information between its fellow observers, and in doing so can synthesize different sources of information embodied in environmental stimuli. In this sense, the collective behavior aspect serves as a pre-processing layer. More generally, these interactions between internal models are the source of a distributed, heterogeneous sampling strategy, which can be used as the input for a variety of machine and deep learning models.

## 2.3   Hybrid Connectionist Approach to Biorealism

We use a biologically plausible approach that approximates the relationship between genotype ($G$) and phenotype ($P$). Rather than the brain, we take inspiration from population genetics to represent the contributions of information encoded in genes ($g$), the temporal expression of these genes ($t$), environmental influences ($e$), and stochastic contributions ($s$) to represent a behavioral phenotype. Stochastic contributions refer to biological noise that in nature can serve both adaptive and diversification functions [27]. We describe further details of a graph-based G-to-P model in the Appendix. Rather than using a black box nervous system which processes environmental information, we use a hybrid genetic representation that combines endogenous genetic contributions with environmental input and feedback to process information and produce a phenotype (see Appendix, Figure 1, Part A). Both the simple and complex epistatic models represent an internal model, which is unique to each observer in the population. This allows us to emphasize the uniqueness of individuals in a population. This is particularly true for the complex model, which can be enabled by even a small population of clones with the same initial genotype.

## 2.4   Epistasis in G-to-P Representations

To understand how the G-to-P mapping operates and transforms the innate components of a genetic representation to an observer capable of perceptual diversity, we use the concept of epistasis [28-30]. Epistasis is enabled in our model by using a deep connectionist architecture (further explained in the Appendix). Each layer enables the recombination of information from our G-to-P representation to the behavioral phenotype. A simple epistatic model [17] consists of a set of genes, a series of hidden layers, and a phenotype. Nodes in the internal layers can represent a number of biological phenomena, such as stochastic noise sources and various regulatory mechanisms. We can also implement a complex epistatic model, which includes a more detailed internal model with a modular structure. In both cases, we leverage the depth of such networks to implement the complexity and richness of the behavioral repertoire.

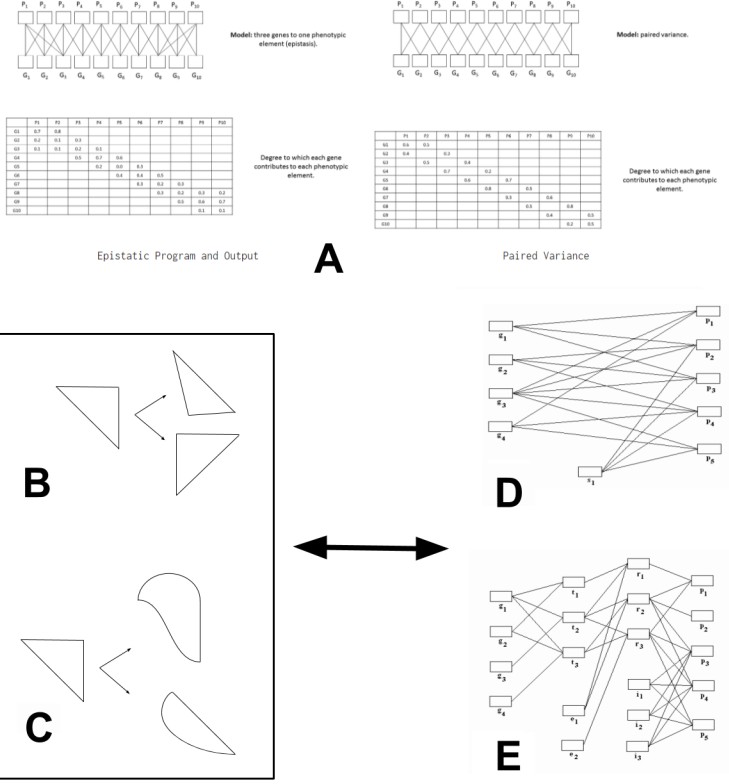

Figure 1: A composite diagram of the G-to-P model and a biologically-inspired observer. A: Statistical summary of the observer phenotype, B: stimulus transformation through affine perspective, C: stimulus transformation through deformed perspective, D: cartoon example of a simple epistatic model, E: cartoon example of a complex epistatic model.

## 2.5 Interactions Between Internal Models

This internal model is an approximation of the multitude effects that arise from genetics, physiology, and environment. Across a population of observers, we can now examine the interactions between observers and among various stimuli. These interactions constitute the environment in which the internal models operate. This allows us to approximate a number of effects, including preference-based selection, which suppresses similar but less-preferred stimuli, perceptual biases such as an aversion to selected stimulus features, and non-realistic object physics, and the attraction-repulsion behavior of other observers. For example, the relative size of virtual objects to avatar body size can produce interpretive illusions in terms of spatial distances [31]. Computational observers may behave similarly, and so verification and communication with other observers might provide interpretive homogenization.

While our internal model can potentially represent great population-level diversity, we also need a method to measure this diversity in a way that translates to the study of behavior. Measures of diversity include applications of mutual information, or a more contextual description based on the soft (fuzzy) classification to deal with vagueness and composition [32]. These measures of diversity provide a statistical summary of the phenotype (as in Appendix, Figure 1, Part A), and provide a population-level parameter with which to measure the behavior of our observer-dependent collective.

Our last expectation is that the phenomenon of cumulative culture [33, 34] can also serve as a guide to explaining behaviors exhibited by our observers. In a typical collective behavior model, there is some form of communication between the observers as they align their behavior. While coordination (and in turn communication) is non-existent among our observers, cumulative behavior nevertheless provides a means for convergent behavior.

# 3    Discussion

We have introduced an architecture for a population of observers, each of which consists of a computational observer represented as a G-to-P mapping. Each of these computational observers carries a genotype that can mutate and observe an environment (operationalized as a set of data points) from multiple perspectives. The individualized frames of reference our observers produce align with what is expected from a spatially embodied system [35]. These perspectives can be shared, and this comparison may lead to an objective representation of the data. As such, our architecture can be used as a pre-processing layer or pre-training model for highly-structured datasets. We can also use this approach to learn more about how artificial observers construct representations of the world. As observers of the world, our diverse population of observers can construct a collective self-regulating model of behavior [36] which is not possible with network architectures that do not consider the role of an observer. In conclusion, the balance of bio-inspiration and contextual integration presented here might serve to enable flexible and adaptive training that maximizes the degree of model robustness while also minimizing the need for model supervision. This may provide fruitful exploration towards generalizability in artificial agents, including expansion upon the train-test regime of learning though more life-like contextual embodiment.

# Appendix

## Implementation of Differential Stimulus Perspectives

Affine perspectives (see Appendix, Figure 1, Part B) are those where the shape is shifted in a way that preserves the ratio between shape components. For example, for a series of points and lines, the resulting shape preserves the original relational information. By contrast, deformed perspectives (see Appendix, Figure 1, Part C) are a distortion of the original image where the ratio between shape components is modified. For a series of points and lines, this could be represented by using parametric curves rather than straight lines. Population of observers with different perceptual histories but same innate traits results in overlapping populations of perspectives.

## Description of Graph-based G-to-P Architecture

The representation of each observer's internal model is an alternative to modeling a mammalian brain. The G-to-P representation is an analogy of a biological learning machine, and is inspired by genetics, biological regulation, and the biological bases of behavior. In particular, we are inspired by a biologically-inspired approach to intelligence that approximates the innateness underlying animal behaviors. From a technical standpoint, the G-to-P mapping consists of a hybrid connectionist/biologically-inspired model. As such, the G-to-P representation is an inverse mapping of specific traits to a genomic and molecular basis. This allows us to encode innate traits that are heritable and can be evolved using a genetic algorithm or other model of selection. The genome (input layer) of the G-to-P representation is customizable to specific problems, and are determined in the individual by introducing mutations and recombination to generate alternate forms. This enables variable phenotypic response to a common stimulus.

The basic genetic representation is a bipartite graph that estimates the genotype to phenotype relationship, or G $\rightarrow$ P. We call this the *simple epistatic model* (see Appendix, Figure 1, Part D). The interaction matrix $W_{GP}$ represents a non-reversible mapping. These elements can be weighted by parameters e and s to account for uniform influences of environment and stochastic effects. Both of these parameters are summary representations of feedback from the environment. When values for $e$ and $s$ approach 1.0, the G $\rightarrow$ P relation can partially recover the inverse mapping P $\rightarrow$ G (phenotype to genotype).

The more complex representation involves the use of deep connectionist models (graphs with many layers). We refer to this as the *complex epistatic model* (see Appendix, Figure 1, Part E), and can be compared performance-wise to the simple model. As this model has many layers, recovering the inverse mapping (P $\rightarrow$ G) is impossible. However, the representation of parameters $e$ and $s$ are more detailed, and consist of a heterogeneous set of units (a distinct layer) in the model.

### Notes on G-to-P Epistasis

The modulatory effects of the environment on gene regulation can be modeled as a series of recurrent connections. For example, we might ask how visual and haptic features modulate the expression of genes and ultimately internal biological complexity. Note that this feedback is limited to the regulatory layers, as recurrent sensory information does not affect the actual structure of the genetic loci and their associated information. The hidden (regulatory) layers also allow for the tuning of the behavioral repertoire in the manner similar to a homeostat.

We also use a fitness criterion to evaluate various components of our G-to-P model. In this specific instance of the G-to-P model, fitness is used at the population level to evaluate the contributions of all observers making a set of observations. All observers contribute to their own point-of-view, but not all observers contribute equally to the global state (final evaluation). Alternatively, fitness can also be applied to individual units in the internal model, which has the effect to select for various epistatic effects [16].

## Broader Impact

The broader impact of this work largely involves the application of the model to real-world data sets. As a preprocessing layer or pretrained model, this representation serves to condition the data in a number of different ways. The ability to discover the potential for context and hidden structure in a given data set is a positive feature of this work, although a potential user might mistake this for an automated method for making interpretations about the data. Another caveat involves our conception of the work "objective", which is not meant to be a statement of non-bias, but rather in opposition to subjectivity.

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
