# OpenReview forum: "Observer-dependent Collective Behavior For Biologically-inspired Processing Models"
_NeurIPS.cc/2020/Workshop/SVRHM — Submitted to SVRHM@NeurIPS_

### Official Review · AnonReviewer3 · 2020-10-27
**Opaque and/or void**

**Rating:** 1
**Confidence:** 4

**Review:**

This reviewer cannot understand the contributions made by the paper. Since this reviewer is a generally educated audience (not necessarily expert in the, uncomprehended, main field of inquiry of this paper) and likely typical of a workshop attendee, it is probable that the paper is not suitable for discussion in this workshop.


To be more specific, first, what are the concrete results? There seems to be no theoretical results, by the conventional criterion of proven mathematical theorems. There are also no empirical results---no specific tasks are considered, no models are implemented, and no performance is reported.

The main concrete contribution of the paper seems to be the proposal of an "epistatic model" or "G-to-P mapping." Since this model looks structurally very similar to a neural network, the authors should have explained the relations between the two (similarity and differences, etc.). As it stands, this contribution is of unclear novelty.


Second, independent of actual results, the writing of this paper is vague. Many key terms are left undefined, statements are overly broad; as a result, the paper is very difficult to follow or comprehend.

For example, what a "phenotype" represents is not explicitly specified. The phenotype is "behavioral" (lines 77, 117, 132) and "encounters sensory information" (line 103); but what on earth IS it? The text refers to the appendix, which does not explain what a phenotype is, either.

For another example, the term "epistatic" is introduced on line 123. It is not explained until the next section (2.4), and there the term is only described but not defined. It would appear that "epistatic" in this context refers to any model that considers more than one "gene" somewhere in the model, e.g., a linear combination of two genes as in Fig 2A, right column. By this (inferred) definition, the epistatic models are still not distinct from artificial neural networks.

As third example, the paper frequently alludes to a "global," "objective," "cumulative" evaluation as the goal and motivation of having a population of individual models with different perspectives (e.g., lines 87--88, 93--97, 218--219). However, how a collective picture may emerge from individual epistatic models is not explained at all.


Finally, the topic of this paper seems to not directly relate to vision, visual representations, or sharing of representations between humans and machines.


In summary, the report is too rough of a sketch for meaningful discussion in this venue.


Minor comments
- Lines 178--180: Affine transformation do NOT preserve ratio between shapes (consider skew for example)
- Lines 190--192: This description of the G-to-P mapping is neither technical (not specific enough for someone else to reproduce) nor "inverse" (isn't it a G-to-P mapping and not P-to-G?)
- What are i and r in Fig 1E?

---

### Official Review · AnonReviewer2 · 2020-10-31
**The work introduces a model for achieving a versatile collective behavior as a data collection mechanism**

**Rating:** 5
**Confidence:** 1

**Review:**

The work discusses a design of an observer that allows to achieve a realistically versatile set of behaviors of a population of observers. The model is inspired by biological models.
While I am not an expert in the area, one significant missing piece in this work for me is the absence of validations for the claims in the paper. The work seems to be purely theoretical. While the model is claimed to hold multiple good and important properties, I could not find any experimental results validating the proposed design. I would recommend to conduct any experiment on an existing well-studied model, e.g., within an environment for reinforcement learning, and provide the results of the experiment with the proposed model.

---

### Official Review · AnonReviewer1 · 2020-11-02
**An Interesting Proposal but Quite Hard to Decipher**

**Rating:** 4
**Confidence:** 2

**Review:**

I don't think I fully understand this paper since it's quite hard to decipher. But I will try to share some useful opinions:

The central proposal of this paper seems quite straight forward: A biological learning agent should have genotype(G)-phenotype(P) differences. As a result, the samples acquired by each different learning agent would be different due to its G-P. Or in other words, each agent would 'see' (perspective) the world differently due to its innate property and its experiences. This is related to the augmentation used in machine learning algorithms but in a heterogeneous fashion. For a typical machine learning algorithm, there is usually a global learning algorithm, and the augmentation is homogeneous. In this proposal, each of the agents would augment its samples from the environment with a different geometric perspective. In collection, the agents' different perspectives may together lead to some interesting behaviors and understanding of the whole environment, etc.

I found this proposal quite interesting!

However, I also found the description of the proposal lacks concrete material as support. Though I understand this is a workshop paper, this fact still bothers me here and there. And I frequently find the description overall general to a level, which is quite hollow. The authors are quite knowledgeable about many different areas and try to make many interesting connections to them. However, as a novice reader, I would appreciate that if the authors can formulate the problem in a clear fashion and demonstrate some concrete application of the idea. The current version as a short paper has too many repetitive general statement and way too many unnecessary technical terms.

Anyway, I think the overall proposal is inspiring and it just needs further clear formulation and solid support.

---

### Decision · Program_Chairs · 2020-11-02

Reject